# Are the Consumption Patterns of Sports Supplements Similar among Spanish Mountain Runners?

**DOI:** 10.3390/nu15020262

**Published:** 2023-01-04

**Authors:** Rubén Jiménez-Alfageme, Noelia Rubio-Quintanilla, David Romero-García, Antonio Jesús Sanchez-Oliver, Isabel Sospedra, José Miguel Martínez-Sanz

**Affiliations:** 1Faculty of Health Sciences, University of Alicante, 03690 Alicante, Spain; 2Food and Nutrition Research Group (ALINUT), University of Alicante, 03690 Alicante, Spain; 3Physiotherapy Department, Faculty of Health Sciences, European University of Gasteiz—EUNEIZ, 01013 Vitoria-Gasteiz, Spain; 4Departamento de Motricidad Humana y Rendimiento Deportivo, Facultad de Ciencias de la Educación, Universidad de Sevilla, 41004 Sevilla, Spain; 5Nursing Department, Faculty of Health Sciences, University of Alicante, 03690 Alicante, Spain

**Keywords:** sports supplements, mountain running, sport nutrition, scientific evidence, performance

## Abstract

Background: The use of sports supplements (SS) to improve sports performance is widespread in all types of athletes, however, the specific characteristics of mountain races may require the use of certain SS. Despite being a sport where the consumption of SS seems widespread, few studies have been conducted in this regard. The objective of this study is to analyze the pattern of SS consumption of mountain runners in relation to the degree of scientific evidence, sex, and level of competition. Methods: Descriptive and cross-sectional study on the consumption and habitual use of SS of 357 federated mountain runners in Spain. Data were collected through a validated questionnaire. Results: From the total sample, 93.84% of the athletes stated that they consumed SS, with no differences observed based on the competitive level or in terms of sex; however, there were significant differences according to the competitive level in terms of the number of SS consumed, with consumption being greater at a higher competitive level (*p* = 0.009). The most consumed SS were sports bars (66.1%), sports drinks (60.5%), sports gels (52.9%), and caffeine (46.2%). Conclusions: The consumption of SS in mountain races is high, and the number of SS consumed is higher as the competition level increases. The four SS most consumed by the participants in this study were all included in category A in the classification of the Australian Institute of Sport (AIS), this category is the one with the greatest scientific evidence.

## 1. Introduction

The sport of mountain running (MR) is becoming increasingly popular, with the number of athletes and competitions increasing in recent years [1]. Since 2001, it has been one of the activities regulated by the Spanish Federation of Mountain and Climbing Sports (FEDME) [2,3]. MR competitions can be held in different scenarios, from low to medium and high mountains. It consists of completing a route along trails or virgin areas with their respective slopes and technical difficulties, in the shortest possible time and by respecting the natural environment. There are different modalities within MR with distances ranging from no more than 21 km to more than 200 km [3]. Race distance will be a factor to be taken into account by athletes, especially when deciding the correct supply of food and hydration during competition [4,5].

It is a demanding sport, where speed is not the only variable to consider, since resistance or aerobic capacity becomes important when competing for long periods of time [6,7]. One of the characteristics of sports practice is the appearance of fatigue, normally associated with a decrease in muscle glycogen and blood glucose [8]. These substrates are essential and can determine the success or failure of the athlete in a test or race. In addition, it is necessary to highlight other factors such as carbohydrate (CHO) intake, hydration, adverse results related to nutrition (e.g., gastrointestinal complaints), and the athlete’s acclimatization [9,10,11,12]. The physiological and psychophysiological characteristics that are important for success may include: aerobic capacity and lactate clearance capacity; career economy; skills characteristics of the mountain race (going up, going down, handling obstacles…); pacing strategies; exogenous and endogenous energy substrate availability and utilization kinetics; thermoregulatory response; gastrointestinal integrity; and functional responses [7,12].

These factors must be taken into consideration when planning training and competitions at a dietary-nutritional level, and for this, it will be necessary to consider the use of sports supplements (SS) on many occasions to ensure the optimal performance of the athlete, as only ensuring it through normal food intake can be tricky, especially when it comes to races that last for long periods of time [4,7]. SS can be defined as “foods, food components, nutrients or non-food compounds that are intentionally ingested in addition to the diet normally consumed with the aim of achieving a specific health and/or performance benefit” [13]. SS are classified by the Australian Institute of Sport (AIS) “ABCD System” according to the level of scientific evidence available to date [14]: (A) Compatible with its use in specific situations in sport, using protocols based on evidence. (B) Those that need additional investigation and that should be used under investigation. (C) Those with little significant evidence of beneficial effects. (D) Substances that are prohibited or at high risk of contamination with substances that could lead to a positive drug test. The use of SS is widespread in all types of sports, but even more so in MR due to the long duration of the competitions, where runners compete anywhere from 3–4 to 24 h [1]. Athletes carry out this type of competition in a self-sufficient or semi-self-sufficient regime, that is, they must bring their minimum requirements of food and hydration, in some cases with the help of the supplies provided by the organizers; this is one of the reasons why the use of SS would be beneficial, as they optimize space and weight in their sports equipment [1]. The training sessions are also usually in a self-sufficiency or semi-sufficiency regime depending on the duration and if the athlete has external help [1,4].

Previous research carried out on the general consumption of SS in different sports showed that between 30 and 100% of athletes consumed SS [15,16,17,18,19,20,21,22]. These values vary depending on the sports discipline, as well as the level of the athlete. Among the most consumed SS, we find isotonic or sports drinks, sports bars, whey protein, iron, vitamin complexes, caffeine, creatine monohydrate, branched-chain amino acids (BCAAs), glutamine, and melatonin [15,16,17,18,19,20,21,22].

To date, few studies have been conducted to determine the prevalence and consumption patterns of SS of mountain runners and have used a small sample as the study population [23]. Although some studies have been carried out focused on nutritional intake of these athletes during a competition [22,24], these studies have focused exclusively on the consumption of food, beverages, and SS during a competition, and not on the general pattern of consumption of SS. Thus, the objective of this study is to analyze the general pattern of consumption, as well as the selection of SS of mountain runners in relation to the degree of scientific evidence, sex, and competitive level. Regarding the main hypotheses of this work, the following are proposed:

**Hypothesis** **1** **(H1).**
*It is expected that the type of supplements most consumed by women also coincide with those consumed by men.*


**Hypothesis** **2** **(H2).**
*The total consumption of SS by women is expected to be lower than by men.*


**Hypothesis** **3** **(H3).**
*Differences are not expected in the consumption and type of SS between mountain runners and other endurance athletes such as triathletes, swimmers, and runners.*


## 2. Materials and Methods

### 2.1. Type of Study

A descriptive and cross-sectional study on the consumption and regular use of SS by mountain runners. The sample size calculation was performed with Rstudio software (version 3.15.0, Rstudio Inc., Boston, MA, USA). The significance level was set a priori at *p* = 0.05. The standard deviation (SD) was set according to the total SS data from previous studies on elite Spanish athletes (SD = 2.1) [25]. With an estimated error (d) of 0.22, the sample size needed was 357 subjects. The study population was selected by non-probabilistic, non-injury, convenience sampling, among sports federations and clubs with a mountain running category throughout Spain.

### 2.2. Participants and Study Sample

The participants were 357 mountain runners aged 37.27 ± 10.52 years old, 262 men and 95 women, of legal age, who had taken part in regional, national, and international competitions for at least two years, with none of the runners suffering from any injuries or illnesses in the 6 months prior to the survey. The competitive level of the participants differed between regional (competitions at the provincial and regional level), national (competitions throughout Spain), and international (competitions worldwide). Table 1 describes the age, basic anthropometric characteristics, and years of sports experience of the study sample.

### 2.3. Instruments

The resource used during the study was based on a questionnaire that was previously used in similar studies [15,17,26]. The selected supplement consumption questionnaire was validated based on content, applicability, structure, and presentation in athletes [27]. The questionnaire contained a total of 36 questions divided into three main sections: the first collected the anthropometric (e.g., age, weight, height, …), personal (e.g., sex), and social data (e.g., an autonomous community of residence) of the respondent (8 questions); the second covered the practice of sports and its context (10 questions, e.g., years of practice, number of competitions…); and the last and most extensive was related to the consumption of SS (18 questions), this part included, among other questions: what supplements they consume, why they consume them, who advises them, where they buy them, when they take them or their perception of results after consumption); it can be found in: Estudio del consumo de suplementos nutricionales en corredores por montaña (Appendix A).

### 2.4. Procedures

To select the study sample, representatives from the national federation (FEDME), and regional mountain and climbing federations in Spain, were contacted via email to inform them about the characteristics of the study and to request their collaboration. After agreeing to participate, an e-mail was sent to them containing the link to the SS consumption questionnaire, so that the mountain runners could complete it voluntarily, electronically, and anonymously. The protocol complied with the Declaration of Helsinki for human research and was approved by the University of Alicante ethics committee with File number UA-2021-02-01.

### 2.5. Statistical Analysis

To verify whether the variables had a normal distribution, a Kolmogorov-Smirnov test was applied, and Levene’s test was used to verify homoscedasticity. The quantitative data obtained were presented as mean (M) + SD, while the qualitative variables were expressed as percentages and frequencies. A two-way ANOVA was performed for the sex factor (male-female), and level of competition (regional, national, and international), to analyze the differences in the total consumption of SS, as well as the SS consumed from the different categories determined by the AIS [14]. Regarding the analysis of the athletes who consumed SS, the reason for consumption, the place where they obtained them, and who advised them to consume them, a chi-square (X2) test was used to verify the existence or not of differences between athletes of different sex and level of competition. As for the SS that were consumed by at least 10% of the sample, an X2 test was performed to verify possible differences according to sex or level of competition. For those variables in which significant differences were found, a pairwise comparison was made using the Bonferroni correction for multiple comparisons. The level of statistical significance was established as *p* < 0.05. The statistical analysis was carried out with the Statistical Package for Social Sciences v.20 software for Windows (SPSS) (IBM, Armonk, NY, USA).

## 3. Results

### 3.1. General Consumption of Sports Supplements and Information Sources

From the total sample, 93.84% of the athletes stated that they consumed SS, with no significant differences being observed depending on the competitive level (92.89% regional level, 95.88% national level, and 95.24% international level), nor according to sex (94.65% of men vs. 91.57% of women). However, significant differences were found according to the competitive level for the total sports supplements (F = 4.741; *p* = 0.009), for the total Group A supplements (F = 5.318; *p* = 0.005), and for performance supplements (F = 9.748; *p* = 0.000) belonging to Group A supplements, with a higher consumption as the competitive level increased. No significant differences were found in terms of sex regarding the number of SS consumed.

Regarding the consumption within each group of supplements determined by the AIS [14], within group A, 94.96% of the sample reported consuming them, with supplements in this group having the highest general consumption. On the other hand, within group B, a consumption of 35.57% of the sample was observed compared to 63.87% from group C, a category with less scientific evidence than the previous one.

Considering sex and according to the competitive level, consumption of 5.47 ± 0.59 sports supplements was observed in women who competed at the regional level, as compared to 6.76 ± 0.34 of men of the same level; nationally, women consumed a mean of 7.52 ± 0.96 supplements while men at the same level consumed 7.69 ± 0.53; and regarding the international level, women reported consuming an average of 7.72 ± 1.38 supplements compared to 9.60 ± 1.45 reported by men of the same level, without significant differences between women and men in any competitive category.

Regarding the answer of the participants on the reason for consumption, the main ones were improvement of performance (66.95%) followed by taking care of health (18.49%), while the rest of the responses referred to alleviating some deficit in their diet, due to a need, or to improve their physical appearance. Regarding the person who motivated the consumption of SS, the dietician-nutritionist (33.61%) stood out as the main person who motivated it, followed by the coach (28.29%), friends (21.57%), and teammates (21%). In addition, the place where athletes usually bought the SS, it was mostly specialized stores (48.18%) and through the internet (44.82%), followed by the pharmacy (15.97%), the mall (13.73%), herbalists (7.56%), and the consultation of the dietician-nutritionist (6.72%).

In relation to the moment in which they used to take the SS, the participants indicated: training and competition days (59.94%), followed by competition days (24.93%), and daily consumption (10.64%). Table 2 shows the mean, standard deviation, median and interquartile of the SS consumed according to the different categories established by the AIS [14], based on sex and level of competition. Table 3 shows the differences between the consumption of SS according to sex, level of competition, and the interaction between both. Regarding sex, significant differences were only found (F = 8.206; *p* = 0.005) in Group C supplements. Finally, on the interaction between sex and level of competition, the only significant difference found was in the consumption of performance supplements (F = 6.314; *p* = 0.002), belonging to the Group A supplements.

### 3.2. Most Consumed Sports Supplements by Sex and Competition Level

Table 4 shows the supplements that were consumed by more than 10% of the sample, according to sex and level of competition. It was observed that sports drinks and bars were consumed the most by the sample (60.5% and 66.1%, respectively), followed by sports gels (52.9%) and caffeine (46.2%). Regarding the differences based on sex, it was found that women had a lower intake than men of sports drinks (*p* = 0.020), CH gainers (*p* = 0.025), and BCAAs (*p* = 0.028); while for iron, the intake of women was higher than that of men (*p* = 0.000). Regarding the level of competition, it was observed that international-level runners had higher intakes of whey protein (*p* = 0.007), iron (*p* = 0.010), vitamin D (*p* = 0.040), and caffeine (*p* = 0.046); while those at the national level had higher intakes of creatine monohydrate (*p* = 0.005) and BCAAs (*p* = 0.015).

Regarding the most consumed SS of each group according to sex, those from group A that were mostly consumed by women were sports bars (72.8%), sports gels (61.05%), sports drinks (57.8%), and caffeine (55.9%); men coincided with women, with sports bars in the first place (63.36%), followed by sports drinks (61.45%), sports gels (50%) and caffeine (42.72%). Regarding the consumption of Group B supplements, it was observed that both men and women coincided with collagen as the most consumed SS in this group (21.05% in women and 18.32% in men). With regard to Group C supplements, differences between the sexes were observed, with magnesium being the most consumed by women (40%), while BCAAs were the most consumed by men (27.10%). Table 5 shows the pairwise comparison between the different categories after the Bonferroni adjustment of the variables measured with significant differences, depending on sex, level of competition, and their interaction. Regarding sex, it was observed that men consumed more Group C supplements than women (*p* = 0.005). As for the level of competition, significant differences were observed when the regional level was compared with the national and international levels (*p* = 0.040–0.001) both in performance supplements and in the total supplements consumed in Group A, with consumption being lower in Group A at the regional level, but not in the case of total sports supplements (*p* = 0.065–0.050). Finally, for the interaction between sex and the level of competition, significant differences were only found when women at the international level were compared with those at the regional and national levels (*p* = 0.009–0.000), with the latter having a lower consumption of performance supplements.

## 4. Discussion

The main objective of this study was to analyze the pattern of SS consumption of MR, including the possible differences according to sex and competitive level of the athletes. The results showed that 93.84% of the sample reported consuming SS. Thus, in comparison to previous studies, a greater consumption of SS was observed than other mountain runners [23], as well as compared to other sports such as fencing (46.9%), sailing (52.4%) or tennis (61.4%) [15,16,18]; nonetheless, it is necessary to mention that the physiological characteristics and nutritional needs of these sports are different [4,13]. However, these findings were only surpassed by rowers, in which 100% of the sample stated that they consumed at least one SS [17]. In addition, a difference in consumption was found between mountain runners of different competitive levels, with the number of SS consumed being higher in athletes from a higher competitive level, with these results similar to those previously found in studies with different Spanish athletes [19,20,23,26]. Although there are very few previous studies that analyzed SS consumption specifically in endurance sports [20,21,23], this is one of the first to do so in mountain runners with a representative sample of this population group.

Studies indicate that the most common reason for consuming SS was to seek an improvement in sports performance followed by taking care of one’s health and, in third place, to alleviate some deficit in the diet [15,16,17,18,19,20], this is similar to our results. Regarding the person who motivated the consumption of SS, the results obtained are encouraging, since the main motivator was a dietitian-nutritionist (D-N) (33.61%), which implies better choices by the athletes when choosing to consume one SS or another regarding its level of scientific evidence [28,29,30]. As secondary motivators, we found coaches (28.29%), friends (21.57%), teammates (21%), and physical trainers (18.77%). These results are similar to those reported in previous studies on mountain runners [23] but differ from those obtained in other sports disciplines, in which the main motivators were unqualified individuals such as friends, teammates, coaches, or themselves [15,16,17,18,19,20]. It is important to consider that many athletes decide to use SS as part of their nutritional strategy, advised, most of the time, by unqualified personnel in sports nutrition.

In reference to the place of purchase, the results showed that mountain runners usually acquired SS mainly in specialized stores (48.18%) and through the internet (44.82%), followed by pharmacies (15.97%) and shopping centers (13.73%). These results are in agreement with those obtained in other similar studies on other sports modalities, with specialized stores, pharmacies, and the Internet is the most common purchasing sites [15,16,17,18,19,20,22]. Athletes buy SS in specialized sites to avoid fraud, doping, or contamination problems [13,30].

Regarding the days in which SS were usually consumed, mountain runners reported consuming them mostly both in training and in competitions (59.94%), followed by only the day of the competition (24.93%) and throughout the year (10.64%), results similar to those obtained in other sports such as rowing [17]. This can be explained by the characteristics of the sport itself. When runners compete over long distances, they have training sessions that last more than two hours, in which they will need to provide adequate nutrition and hydration. To do this, and to train the gastrointestinal tract, they use SS, to prevent the appearance of possible gastrointestinal discomfort during competition [31,32,33].

The results of the analysis of the SS groups determined by the AIS [14], according to the level of scientific evidence, showed statistically significant differences within the performance supplements subgroup (group A) in terms of the level of competition (F = 9.748; *p* = 0.000) with the international level showing the highest consumption (0.67 ± 0.05 regional; 0.97 ± 0.09 national level; 1.33 ± 0.31 international level). In this same subgroup, significant differences were also observed regarding the interaction between sex and level of competition (F = 6.314; *p* = 0.002). Statistically significant differences were observed in the total of group A according to the competition level, with a higher consumption observed at the level of international competitions (F = 5.318; *p* = 0.005). These results coincide with recent studies, where significant differences were also observed regarding the level of competition, which leads us to the assumption that the level of the athlete determines the consumption of SS [16,19,34]. In group C, significant differences were observed in terms of sex (F = 8.206; *p* = 0.005), with men having a higher consumption compared to women (1.64 ± 0.12 in men compared to 1.21 ± 0.62 in women), but this difference was not observed according to sex in the total SS or in any other category, which differs from previous studies, which showed that men consumed more than women [16,20,27]. The current evidence indicates that athletes who compete at a lower level are less informed about SS [35], but our study differs from these previous results as the main motivator for the consumption of SS was a D-N without differences in the level of competition.

Regarding the average consumption in each of the AIS groups [14], it was observed that in group A, the average consumption of SS was 4.84 ± 0.16 exceeding the average consumption of groups B (0.52 ± 0.04) and C (1.53 ± 0.10), although it is true that the high consumption within group A was largely due to the high consumption of the SS in the sports food subgroup (3.54 ± 0.12), and not much from the rest of the subgroups included within group A. The high consumption of SS may be due in part to the characteristics of the sports modality itself, in which the supply of carbohydrates during training and competitions is of vital importance. Moreover, these are usually consumed in the form of SS for the comfort and ease of transportation [1,24,36]. It was also observed that the consumption of SS from group C (supplements that have little or no evidence of beneficial effects) was 1.53 ± 0.10, tripling the amount of consumption of SS from group B (supplements that have some benefit, but more research is needed), which was 0.52 ± 0.04. This should be considered when dealing with athletes in nutrition consultations, and it is also an interesting topic to study in nutritional education programs [28,29,30].

The results from the present study indicated that the four most consumed SS by mountain runners were sports bars, sports drinks, sports gels, and caffeine. These results are similar to those obtained in previous research, in which, in general, the most consumed SS were isotonic or sports drinks, energy bars, and multivitamins [15,16,17,18,19,20,22]. It should be noted that the four most consumed supplements are classified within group A of the AIS classification with the highest degree of current scientific evidence [14].

If we observe the results according to sex, the four most consumed SS coincided in both sexes. In general, mountain runners are aware of the importance of carbohydrate and liquid consumption during this type of event. Specifically, the three most used SS were supplements, which are usually used in this type of competition due to their ease of transport and tolerance, as well as their provision of carbohydrates [37,38,39,40]. In general, these results coincide with those reported in other studies, in which the most consumed SS were sports bars, isotonic or sports drinks, and multivitamins, with no statistically significant differences between sexes [15,16,17,18,19,20,22]. The consumption of this type of SS, such as sports drinks, sports gels, sports bars, and sports confectionery groups can be used before or during training and/or competition, while the protein powders, protein bars, and mixed macronutrient supplement groups can be used after training and/or competition, and their consumption can contribute to reaching recommended nutrient intakes [13,41,42]. The current nutritional recommendations for sports events (training and competition) state that during physical activity, whilst an event is taking place, the average hourly intake should be 500 mL of liquids, 300–600 mg of sodium, and 60–120 g of carbohydrates [4,41,42]. The athlete’s nutritional needs during exercise could be met with the intake of water, sports drinks, sports gels, sports bars, and food. In addition, post-exercise nutritional recovery is performed through the intake of carbohydrates and proteins (0.8 g carbohydrate/kg body weight plus 0.2–0.4 g protein/kg body weight) [4,41].

There are many factors that can influence the success or failure of mountain runners [6,7,8,10,11,12]. The physiological characteristics of the MR explain the choice of the most consumed SS, i.e., the use of energy substrates by the body during these tests, which the athletes will have to provide exogenously in order to obtain the desired performance to carry out the competition [1,37,38,39,40]. The bars and gels provide large amounts of carbohydrates in a small amount of product, which in addition to producing less gastrointestinal discomfort, allows the athletes to optimize the space they have in their means of food transport (backpacks, belts, pockets.) [1,4,24,36,38]. Another important reason that can explain the consumption of sports drinks is thermoregulation [43,44,45,46]. Especially when the race takes place in hot environmental conditions, there is an increase in the production of internal body heat that must be controlled through different strategies (acclimatization, hydration…) to maintain body homeostasis, which will allow the athlete to have greater success [43,44,45,46,47]. This is why athletes usually choose replenishment drinks to hydrate themselves, as dehydration appears in up to 20.4% of these competitions. CHO and Na are also provided in these drinks, which allow them to maintain a good state of body homeostasis, to prevent hyponatremia associated with exercise, which is observed in up to 10.3% of the participants [48,49]. As with all foods, gut training is necessary so that athletes can tolerate the recommended intake of carbohydrates, sodium, and liquids during the training sessions or competition periods, as well as to decrease gastrointestinal problems [49,50], as around 9.4% of the participants suffer from them in this type of competition.

Although it is true that the results of this study showed that most of the SS consumed by mountain runners were classified within category A of the AIS [14], it was also observed that there was still a fairly broad consumption of supplements with little or no scientific evidence in this regard (Groups B and C). This is similar to another recent study carried out in an endurance sport modality in open water swimmers [51]. Therefore, it is highly important for athletes to be educated by nutrition professionals, so they can make better SS choices by choosing safe and effective supplements.

This research has some limitations that must be mentioned to improve its applicability in real sports contexts. Although the tool to evaluate SS consumption in athletes was a validated questionnaire, the collection of this information in a self-reported and retrospective manner can induce errors in the number and type of SS consumed. However, it is one of the first studies to have a significant sample in this type of population. As future research lines, efforts can be made to collaborate with federations in other countries and thus have a representative SS consumption pattern worldwide and check whether the SS consumption is similar in all of them according to the competitive level and sex.

## 5. Conclusions

The consumption of SS in mountain races is high due to their physiological characteristics, with this consumption being higher as the competitive level of the athletes increases. The most consumed SS by mountain runners were sports bars, replacement drinks, sports gels, and caffeine, all included in category A, the group with the most scientific evidence to date. Similarly, a large part of the athletes obtained advice from a D-N, which would explain why most of the SS consumed by athletes were part of group A. In general, the results of this work were similar to other works carried out in other sports modalities.

## Figures and Tables

**Table 1 nutrients-15-00262-t001:** Descriptive data of the mountain runners at the regional, national, and international levels.

Level of Competition	Sex	Age (Years)	Height (cm) *	Body Mass (kg) *	BMI (kg·m^−2^) *	Exp. (Years)
Regional (n = 239)	M (n = 178)	37.88 ± 10.30	175.33 ± 6.21	72.03 ± 9.11	23.42 ± 2.55	5.17 ± 3.01
	F (n = 61)	37.20 ± 9.40	162.70 ± 5.66	56.04 ± 5.99	21.16 ± 1.93	4.36 ± 3.03
National (n = 97)	M (n = 74)	38.23 ± 11.15	175.86 ± 5.45	70.07 ± 8.29	22.65 ± 2.42	4.84 ± 2.76
	F (n = 23)	36.13 ± 9.79	163.26 ± 6.50	55.37 ± 5.31	20.79 ± 1.84	4.30 ± 3.14
International (n = 21)	M (n = 10)	30.8 ± 7.77	178 ± 8.19	69.8 ± 8.83	21.95 ± 1.79	3.9 ± 2.08
	F (n = 11)	29.45 ± 8.51	166 ± 5.35	54.36 ± 6.02	19.67 ± 1.24	2.82 ± 1.60

* Self-reported height and weight. BMI was calculated with self-reported height and body mass. (Exp: experience, M: male, F: female).

**Table 2 nutrients-15-00262-t002:** Descriptive data of the SS consumed in the different categories defined by the AIS [14], as a function of sex and level of competition.

Variable		Sex		Level of Competition							
		M	F	R	N	I			Total		
		Mean ± SD	Mean ± SD	Mean ± SD	Mean ± SD	Mean ± SD	Med.	IQ	Mean ± SD	Med.	IQ
Total SS		7.13 ± 0.29	6.23 ± 0.45	6.43 ± 0.30	7.65 ± 0.45	8.62 ± 1.00	8.00	6	6.89 ± 0.25	6.00	6
Group A	Sports Food	3.68 ± 0.14	3.15 ± 0.23	3.37 ± 0.15	3.90 ± 0.23	3.81 ± 0.46	4.00	2	3.54 ± 0.12	3.00	3
	Medical Supplement	0.46 ± 0.05	0.66 ± 0.10	0.49 ± 0.06	0.49 ± 0.09	0.90 ± 0.23	1.00	2	0.52 ± 0.05	0.00	1
	Performance supplement	0.83 ± 0.05	0.66 ± 0.10	0.67 ± 0.05	0.97 ± 0.09	1.33 ± 0.31	1.00	2	0.79 ± 0.05	1.00	1
	Total	4.97 ± 0.18	4.47 ± 0.33	4.52 ± 0.20	5.36 ± 0.30	6.05 ± 0.66	6.00	5	4.84 ± 0.16	5.00	4
Group B		0.51 ± 0.05	0.55 ± 0.08	0.48 ± 0.05	0.60 ± 0.09	0.57 ± 0.19	0.00	1	0.52 ± 0.04	0.00	1
Group C		1.64 ± 0.12	1.21 ± 0.62	1.43 ± 0.12	1.67 ± 0.19	2.00 ± 0.47	1.00	1	1.53 ± 0.10	1.00	2
Group D		0.01 ± 0.01	0.00 ± 0.00	0.00 ± 0.00	0.02 ± 0.02	0.00 ± 0.00	0.00	0	0.01 ± 0.01	0.00	0

SS: sports supplements; SD: standard deviation; M: male; F: female; R: regional; N: national; I: international.

**Table 3 nutrients-15-00262-t003:** ANOVA of the SS consumed in the different categories defined by the AIS [14], as a function of sex, level of competition their interaction.

Variable		Model											
		Sex		Level of Competition		Sex–Level of Competition (Mean ± SD)							
		F	*p*	F	*p*	R		N		I		F	*p*
						M	F	M	F	M	F		
Total SS		1.939	0.165	4.741	0.009	6.76 ± 0.34	5.47 ± 0.59	7.69 ± 0.53	7.52 ± 0.96	9.60 ± 1.45	7.72 ± 1.38	0.466	0.628
Group A	Sports Food	1.850	0.175	2.978	0.052	3.56 ± 0.17	2.80 ± 0.29	3.88 ± 0.27	3.96 ± 0.48	4.30 ± 0.72	3.36 ± 0.69	0.925	0.398
	Medical Supplement	1.007	0.316	1.738	0.177	0.43 ± 0.07	0.66 ± 0.11	0.49 ± 0.10	0.52 ± 0.18	0.80 ± 0.28	1.00 ± 0.26	0.292	0.747
	Performance supplement	1.243	0.266	9.748	0.000	0.76 ± 0.06	0.38 ± 0.11	1.00 ± 0.10	0.87 ± 0.18	0.80 ± 0.27	1.82 ± 0.26	6.314	0.002
	Total	0.175	0.676	5.318	0.005	4.76 ± 0.23	3.84 ± 0.39	5.36 ± 0.35	5.35 ± 0.63	5.90 ± 0.96	6.18 ± 0.91	0.806	0.448
Group B		0.082	0.775	1.070	0.344	0.48 ± 0.06	0.47 ± 0.11	0.55 ± 0.10	0.74 ± 0.17	0.60 ± 0.26	0.54 ± 0.25	0.381	0.684
Group C		8.206	0.005	1.571	0.209	1.52 ± 0.14	1.16 ± 0.24	1.74 ± 0.22	1.43 ± 0.39	3.10 ± 0.59	1.00 ± 0.57	2.130	0.120
Group D		0.241	0.624	0.423	0.655	0.00 ± 0.01	0.00 ± 0.01	0.03 ± 0.01	0.00 ± 0.02	0.00 ± 0.03	0.00 ± 0.03	0.423	0.655

SS: sports supplements; SD: standard deviation; M: male; F: female; R: regional; N: national; I: international.

**Table 4 nutrients-15-00262-t004:** Distribution (%) of the most consumed supplements as a function of sex and level of competition according to the categories defined by the AIS [14].

Category		Supplement	Total (%)	Sex (%)			Level of Competition (%)			
				M	F	*p*	R	N	I	*p*
Group A	Sport Foods	Sports drinks	60.5	64.1	50.5	0.020	57.7	70.1	47.6	0.051
		Recovery shakes	15.7	11.6	17.2	0.199	14.6	16.5	23.8	0.524
		Sports gel	52.9	48.4	54.6	0.303	53.1	51.5	57.1	0.892
		Sports confectionery	16.0	17.6	11.6	0.173	15.5	15.5	23.8	0.600
		Sports bars	66.1	65.6	67.4	0.762	66.9	64.9	61.9	0.861
		Protein bar	29.1	27.9	32.6	0.381	29.7	30.9	14.3	0.296
		Electrolytes	28.6	29.0	27.4	0.762	25.9	32.0	42.9	0.178
		Whey protein	24.9	26.7	20.0	0.195	20.1	33.0	42.9	0.007
		CHO Gainers	21.8	24.8	13.7	0.025	21.3	24.7	14.3	0.545
		Maltodextrin	15.7	17.9	9.5	0.052	13.4	20.6	19.0	0.233
	Medical Supplements	Iron	12.3	8.0	24.2	0.000	11.3	10.3	33.3	0.010
		Vitamin D	10.9	10.3	12.6	0.534	8.4	14.4	23.8	0.040
		Multivitamin	15.7	15.3	16.8	0.718	15.5	14.4	23.8	0.557
	Performance Supplements	Caffeine	46.2	48.1	41.1	0.238	42.3	51.5	66.7	0.046
		Creatine monohydrate	19.3	20.6	15.8	0.308	14.6	29.9	23.8	0.005
Group B		Antioxidants Vitamin C	13.4	13.7	12.6	0.786	11.7	17.5	14.3	0.365
		Collagen	19.0	19.5	17.9	0.738	19.7	18.6	14.3	0.826
Group C		BCAA	25.2	28.2	16.8	0.028	21.3	36.1	19.0	0.015
		Glutamine	10.9	12.2	7.4	0.195	11.3	11.3	4.8	0.647
		Magnesium	29.1	30.2	26.3	0.481	32.6	19.6	33.3	0.053

M: male; F: female; R: regional; N: national; I: international.

**Table 5 nutrients-15-00262-t005:** Post hoc comparison between the variables with significant differences.

Model	Variable		Comparison According to Groups		Difference in Means ± SD	*p*	95% IC
Sex	Group C		M–F		0.92 ± 0.32	0.005	0.281 to 1.559
Level of Competition		Total SS	R–N		−1.49 ± 0.65	0.065	−3.042 to 0.065
			R–I		−2.55 ± 1.06	0.050	−5.096 to 0.003
			N–I		−1.06 ± 1.14	1.000	−3.809 to 1.693
	Group A	Performance supplements	R–N		−0.36 ± 0.12	0.009	−0.656 to −0.072
			R–I		−0.74 ± 0.20	0.001	−1.218 to −0.259
			N–I		−0.37 ± 0.21	0.249	−0.892 to 0.143
		Total	R–N		−1.06 ± 0.425	0.040	−2.082 to −0.036
			R–I		−1.74 ± 0.70	0.039	−3.423 to −0.064
			N–I		−0.68 ± 0.75	1.000	−2.497 to 1.128
Sex–Level Competition	Group A	Performance supplements	M	R–N	−0.24 ± 0.12	0.152	−0.526 to 0.054
				R–I	−0.04 ± 0.28	1.000	−0.717 to 0.645
				N–I	0.20 ± 0.29	1.000	−0.506 to 0.906
			F	R–N	−0.49 ± 0.21	0.060	−1.000 to 0.015
				R–I	−1.44 ± 0.28	0.000	−2.121 to −0.762
				N–I	−0.95 ± 0.31	0.009	−1.709 to −0.188

SS: sports supplements; SD: standard deviation; M: male; F: female; R: regional; N: national; I: international.

## Data Availability

The data presented in this study are available in the tables of this article. The data presented in this study are available on request from the corresponding author.

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
