# Peer review of "Are the Consumption Patterns of Sports Supplements Similar among Spanish Mountain Runners?"

_nutrients, 2023, doi:10.3390/nu15020262_

Round 1

Reviewer 1 Report

Comments to authors:

You provides some reasonably interesting data on the use of sports supplements by mountain runners. However, I think the reporting of your results could be clearer and there is scope to add some further insight to your discussion.

Title: 'sports supplements' needs writing in full in the title rather than abbreviating to SS.

Abstract: this lacks any data or p values. It would be useful to report some of your key data/findings.

Category A has no meaning at the point the abstract is read. Some people may only read the abstract. This needs considering.

Introduction:  The introduction provides some reasonable context for your study, but I think it would be useful to provide more explanation of the physiological demands of mountain running, likely exercise intensity etc. 

You introduce the AIS system of supplement classification but do not define the categories/ It would be useful to do this in the introduction or methods sections.

You skim over the previous research in this area i.e. refs 22-24. It would be useful to report some detail of what they/you previously found to strengthen the novelty of this study. Also check these references for accuracy - Martinez et al seems to be in the Journal of Sports Sciences not Nutrients as stated in your reference list.

You end with three hypotheses. These seem to swap between null hypotheses (1 and 3) and alternative hypotheses (number 2) and seem a bit of an add on. I wonder whether it would be best just to end the introduction with your study objective?

Line 51: CH seems an odd abbreviation for carbohydrate. The more standard one would be CHO. 'adverse results related to nutrition' is very vague be more specific about what you mean.

Line 54: 'dietary-nutritional' you don't need both words.

Line 62. Can you include some data to illustrate how widespread the use of SS is rather than just stating it is widespread?

Line 63 - competition length here is 3-4 to 24 hours but on line 43 you state MR can be 3km to 200 km. 3 km would not take 3 hours so these lines are not consistent with each other.

Lines 62-68. Do MR athletes also have the same issues re: food during long training sessions? 

Method: 

Line 89:  'regular use of SS of' change second of to 'by'.

Table 1. Weight heading in table should be reported as ' Body mass'

Line 117: was the questionnaire validated for use in mountain runners, other long distance runners or athletes in general. Perhaps this could be stated? Could the questionnaire be supplied as a supplementary file or a link?

Line 132 - 'UA ethics committee' UA needs writing in full because it is not clear what this abbreviation means.

Line 142 and at various other places in the manuscript change 'gender' to 'sex'.

Line 144 - as mentioned above the AIS system needs to be explained somewhere in the intro or methods in more detail.

Statistical analysis - Which post hoc tests did you use?

Results:

I think this whole section could be improved. A number of group comparisons seem to be reported in the text without the associated p values and test statistic. Also the section is quite difficult to follow and the flow could be improved. I wonder whether it would benefit from some subheadings?

Table 2: why report the mean and the median? Use the mean for data that is  normally distributed and the median for the data that is not normally distributed. 

Table 5: 95% CI would normally be placed next to the mean difference and the SD removed. What does the 'a' mean in the 95% CI column between the upper and lower boundaries?

Discussion: Much of this reads like a results section with the inclusion of p values and test statistics. There is not much in the way of insight. 

Lines 263 - 265. You state SS use was higher than previous studies but do not state the % use in those studies so the reader has no idea of the difference in use across sports/previous studies.

Line 271. Are there really only two studies that have analysed SS use in endurance sports? 

Line 274-284. There is some data here that is not in your results. This whole paragraph reads like a results section. Can you add a bit of insight/discussion of your findings?

Line 285-290: Same as above can you add some discussion about sources of supplements?

Line 291 'training and competitions' versus 'day of competition'. This seems a strange comparison. Surely it would have been better to report during training versus during/day of competitions'? Was this an issue with your questionnaire design?

Prevalence of use on day of competition seems very low 24.93% given your arguments regarding race length and having to bring all their own food and drink? Do most MRs not use sports drinks? Or have the respondents not understood the question?

Line 316 - I do not think you can really infer that MRs that compete at a lower level are less informed about SS based on your data, because you did not directly test this.

Does running speed dictate whether runners are more likely to make use of supplements or foods/normal drinks?

Line 387 - you need to acknowledge whether questionnaire was validated for use in your type of population and perhaps discuss any limitations in the scope of questions?

Author Response

You provides some reasonably interesting data on the use of sports supplements by mountain runners. However, I think the reporting of your results could be clearer and there is scope to add some further insight to your discussion.

Title: 'sports supplements' needs writing in full in the title rather than abbreviating to SS.

Response of the authors: According to the reviewer's suggestions, the abbreviating SS in the title has been replaced by “sports supplements”.

Abstract: this lacks any data or p values. It would be useful to report some of your key data/findings.

Response of the authors: According to the reviewer's suggestions, p value of the differences and data (%) of the most consumed SS have been added to the abstract.

Category A has no meaning at the point the abstract is read. Some people may only read the abstract. This needs considering.

Response of the authors: According to the reviewer's suggestions, it has been added an explanation about category A to improve the understanding.

Introduction:  The introduction provides some reasonable context for your study, but I think it would be useful to provide more explanation of the physiological demands of mountain running, likely exercise intensity etc. 

Response of the authors: According to the reviewer's suggestions, a detailed explanation on the main physiological challenges of mountain running has been added to the introduction (second paragraph).

You introduce the AIS system of supplement classification but do not define the categories/ It would be useful to do this in the introduction or methods sections.

Response of the authors: According to the reviewer's suggestions, the definition of the categories of this classification have been included in the introduction (third paragraph).

You skim over the previous research in this area i.e. refs 22-24. It would be useful to report some detail of what they/you previously found to strengthen the novelty of this study. Also check these references for accuracy - Martinez et al seems to be in the Journal of Sports Sciences not Nutrients as stated in your reference list.

Response of the authors: According to the reviewer's suggestions, additional information on these works has been added to reinforce the novelty of our study. The wrong reference has also been corrected.

You end with three hypotheses. These seem to swap between null hypotheses (1 and 3) and alternative hypotheses (number 2) and seem a bit of an add on. I wonder whether it would be best just to end the introduction with your study objective?

Response of the authors: According to the reviewer's suggestions, we appreciate your response, but the hypotheses have been included since it is an evaluable aspect of the quality of the studies when a systematic review and meta-analysis is carried out. Specifically in the STROBE Statement—checklist of items that should be included in reports of observational studies (https://www.strobe-statement.org/download/strobe-checklist-cohort-case-control-and-cross-sectional-studies-combined) item 3 refers to this information.

Line 51: CH seems an odd abbreviation for carbohydrate. The more standard one would be CHO. 'adverse results related to nutrition' is very vague be more specific about what you  mean.

Response of the authors: According to the reviewer's suggestions, the suggested abbreviation has been modified and an example on nutrition related adverse outcomes has been included.

Line 54: 'dietary-nutritional' you don't need both words.

Response of the authors: According to the reviewer's suggestions, we appreciate sincerely your comment, but both words have been used for their definition, “dietary” refers to food/supplements and “nutritional” to nutrients. When sports planning is carried out, it is carried out at both levels. However, we can keep only the “nutritional” word if you deem it appropriate.

Line 62. Can you include some data to illustrate how widespread the use of SS is rather than just stating it is widespread?

Response of the authors: According to the reviewer's suggestions, the required information appears in paragraph 5 of the introduction, and we did not want to duplicate it.

Line 63 - competition length here is 3-4 to 24 hours but on line 43 you state MR can be 3km to 200 km. 3 km would not take 3 hours so these lines are not consistent with each other.

Response of the authors: According to the reviewer's suggestions, erroneous distance value has been replaced.

Lines 62-68. Do MR athletes also have the same issues re: food during long training sessions? 

Response of the authors: According to the reviewer's suggestions, Training sessions are also self-sufficiency or semi-sufficiency depending on the duration and whether the athlete has external help. This information has been added in the introduction.

 Method: Line 89:  'regular use of SS of' change second of to 'by'.

Response of the authors: According to the reviewer's suggestions, the second “of” has been changed to “by”.

Table 1. Weight heading in table should be reported as ' Body mass'

Response of the authors: According to the reviewer's suggestions, “weight” has been replaced by “body mass” in table 1.

Line 117: was the questionnaire validated for use in mountain runners, other long distance runners or athletes in general. Perhaps this could be stated? Could the questionnaire be supplied as a supplementary file or a link?

Response of the authors: According to the reviewer's suggestions, the questionnaire is validated for all types of athletes and it has been added as a supplementary file. In addition, the information on the questions of the same has been expanded.

Line 132 - 'UA ethics committee' UA needs writing in full because it is not clear what this abbreviation means.

Response of the authors: According to the reviewer's suggestions, the full name of UA: University of Alicante has been included.

Line 142 and at various other places in the manuscript change 'gender' to 'sex'.

Response of the authors: According to the reviewer's suggestions, the word “gender” has been replaced by “sex” throughout the manuscript.

Line 144 - as mentioned above the AIS system needs to be explained somewhere in the intro or methods in more detail.

Response of the authors: According to the reviewer's suggestions, the AIS classification has been defined in the introduction.  

Statistical analysis - Which post hoc tests did you use?

Response of the authors: According to the reviewer's suggestions, the post hoc test used was the Bonferroni adjustment. It has been added in the statistical analysis section.

Results: I think this whole section could be improved. A number of group comparisons seem to be reported in the text without the associated p values and test statistic. Also the section is quite difficult to follow and the flow could be improved. I wonder whether it would benefit from some subheadings?

Response of the authors: According to the reviewer's suggestions, some subheadings have been included in the results section to improve understanding.

Table 2: why report the mean and the median? Use the mean for data that is  normally distributed and the median for the data that is not normally distributed. 

Response of the authors: According to the reviewer's suggestions, we appreciate your comment and median information has been removed.

Table 5: 95% CI would normally be placed next to the mean difference and the SD removed. What does the 'a' mean in the 95% CI column between the upper and lower boundaries?

Response of the authors: According to the reviewer's suggestions, the “a” has been changed to “to” in the 95% CI column, it was an error in the translation of the work.

Do you prefer that we indicate the CI next to the mean value?

Discussion: Much of this reads like a results section with the inclusion of p values and test statistics. There is not much in the way of insight. 

Response of the authors: According to the reviewer's suggestions, we do not understand the comment, since we have tried to discuss the results of other studies following the style of other similar publications made by the authors, including statistics or p values for their comparison: (e.g., https://www.mdpi.com/2072-6643/14/24/5211; https://www.mdpi.com/2072-6643/10/10/1341/htm)

Lines 263 - 265. You state SS use was higher than previous studies but do not state the % use in those studies so the reader has no idea of the difference in use across sports/previous studies.

Response of the authors: According to the reviewer's suggestions, the percentages in the cited studies have been added.  

Line 271. Are there really only two studies that have analysed SS use in endurance sports? 

Response of the authors: According to the reviewer's suggestions, the evidence on the consumption of SS available until now in endurance sports is very low. The studies available up to now analyze the consumption of SS in a specific sports modality, although it is true that other studies that analyze it in a heterogeneous group of athletes without separating them by sport, such as the included reference.

Line 274-284. There is some data here that is not in your results. This whole paragraph reads like a results section. Can you add a bit of insight/discussion of your findings?

Response of the authors: According to the reviewer's suggestions, this paragraph has been reformulated.

Line 285-290: Same as above can you add some discussion about sources of supplements?

Response of the authors: According to the reviewer's suggestions, this paragraph has been improved and some discussion has been added.

Line 291 'training and competitions' versus 'day of competition'. This seems a strange comparison. Surely it would have been better to report during training versus during/day of competitions'? Was this an issue with your questionnaire design?

Response of the authors: According to the reviewer's suggestions, “training and competitions” refers to both moments, however, “day of competition” refers only to the consumption of SS in competitions. The questionnaire asks if they consume SS in training, in competition, in both or throughout the year. For what is discussed later, most athletes consume them in both training and competition.

Prevalence of use on day of competition seems very low 24.93% given your arguments regarding race length and having to bring all their own food and drink? Do most MRs not use sports drinks? Or have the respondents not understood the question?

Response of the authors: As explained above, most runners consume them both in training and in competition.

Line 316 - I do not think you can really infer that MRs that compete at a lower level are less informed about SS based on your data, because you did not directly test this.

Response of the authors: According to the reviewer's suggestions, the phrase has been reformulated to improve its understanding.

Does running speed dictate whether runners are more likely to make use of supplements or foods/normal drinks?

Response of the authors: According to the reviewer's suggestions, in this type of competitions, athletes consume food and/or SS given the physiological and nutritional needs already explained. Although we cannot give you an answer with our results, since the running speed of the runners is not known. However, we consider that with greater running speed and effort, the possibility of consuming some types of food and SS is limited.

Line 387 - you need to acknowledge whether questionnaire was validated for use in your type of population and perhaps discuss any limitations in the scope of questions?

Response of the authors: According to the reviewer's suggestions, we appreciate the suggestion, and the limitations paragraph has been improved.

Reviewer 2 Report

You work is well conduct and analyzed. 

Some remarks:

Discussion: it is not right to compare the consumption of mountain runners with fencing sailing or tennis (line 265), it is obvious the the needs are different. This sentence could be removed.

Rather is interesting where the runners acquire SS, and more and more who motivates the use. A dedicated underline is appropriate.  

Author Response

You work is well conduct and analyzed. 

Some remarks:

Discussion: it is not right to compare the consumption of mountain runners with fencing sailing or tennis (line 265), it is obvious the the needs are different. This sentence could be removed.

Response of the authors: According to the reviewer's suggestions, we agree with your comment, however, given the scant literature on supplement use in endurance sports, it has been compared to other sports where the literature is larger. Anyway, a clarification has been added in this paragraph considering your comment.

Rather is interesting where the runners acquire SS, and more and more who motivates the use. A dedicated underline is appropriate.

Response of the authors: According to the reviewer's suggestions, the aspects related to the explanation of the motivator and the place of acquisition of SS have been improved in the discussion section.
